# The Role of Heparin in COVID-19: An Update after Two Years of Pandemics

**DOI:** 10.3390/jcm11113099

**Published:** 2022-05-30

**Authors:** Marco Mangiafico, Andrea Caff, Luca Costanzo

**Affiliations:** 1Unit of Internal Medicine, Policlinico “G. Rodolico—San Marco”, 95100 Catania, Italy; marcomangiafico@hotmail.it (M.M.); caff.andrea@gmail.com (A.C.); 2Unit of Angiology, Department of Cardio-Thoraco-Vascular, Policlinico “G. Rodolico—San Marco” University Hospital, University of Catania, 95100 Catania, Italy

**Keywords:** COVID-19, heparin, low molecular weight heparin, coagulopathy, thromboprophylaxis

## Abstract

Coronavirus disease 2019 (COVID-19) is associated with an increased risk of venous thromboembolism (VTE) and coagulopathy, especially in critically ill patients. Endothelial damage induced by severe acute respiratory syndrome coronavirus 2 (SARS-CoV-2) is emerging as a crucial pathogenetic mechanism for the development of complications in an acute phase of the illness and for several postdischarge sequalae. Heparin has been shown to have a positive impact on COVID-19 due to its anticoagulant function. Moreover, several other biological actions of heparin were postulated: a potential anti-inflammatory and antiviral effect through the main protease (M^pro^) and heparansulfate (HS) binding and a protection from the damage of vascular endothelial cells. In this paper, we reviewed available evidence on heparin treatment in COVID-19 acute illness and chronic sequalae, focusing on the difference between prophylactic and therapeutic dosage.

## 1. Introduction

In December 2019, a novel coronavirus, the SARS-CoV-2, emerged in the city of Wuhan (China) [1] and was responsible for unusual viral pneumonia that has caused cases of acute respiratory distress syndrome [2].

On 30th January 2020, the World Health Organization (WHO) proclaimed the SARS-CoV-2 infection as a public health emergency of international concern [3], that rapidly spread all around the world. After almost two years of the pandemic, on 18 March 2022 WHO reported 464,809,377 confirmed cases of coronavirus disease 2019 (COVID-19), including 6,062,536 deaths [4]. 

The clinical presentation of COVID-19 is quite variable and may vary from asymptomatic or mild respiratory symptoms to pneumonia with respiratory failure and mortality [5]. Furthermore, many patients showed coagulation abnormalities: increased D-dimer concentration upon hospital admission, a decrease in platelet count, and a prolongation of the prothrombin time suggested the presence of a hypercoagulable state in COVID-19 that could lead to an increased risk of thromboembolic complications [6,7]. Indeed, venous thromboembolism (VTE) has emerged as a common complication, particularly in critically ill patients [8,9,10,11,12,13]. A recent study has confirmed the increased prevalence of VTE in critically ill COVID-19 patients both in ante-mortem and post-mortem cohorts and an improvement of prognosis after the change in anticoagulation practice [14]

VTE may also occur after hospital discharge, with up to 80% of events occurring in the post-hospital discharge period (30–45 days) following index hospitalization [15]. 

Therefore, anticoagulation management in COVID-19 represents a therapeutic challenge for clinicians. In this review, we focused on the role of heparin in various COVID 19 clinical settings. Particularly, we reviewed the pathophysiology of vascular damage and the hypercoagulative state related to SARS-CoV-2 infection and the most recent evidence of treatment with heparin considering its pleiotropic and anticoagulant effects in both the acute phase and postdischarge.

## 2. Endothelial Damage in COVID-19

The endothelium is a single layer of endothelial cells (ECs) that constitutes the inner cellular lining of the blood vessels (arteries, veins and capillaries) and the lymphatic system [16]. ECs have different functions that depend on the tissues and organs. The most important function of ECs is to control vascular permeability and regulate vascular tone through the synthesis of several factors [17]. Furthermore, ECs are involved in the adhesion and aggregation of platelets, activation, adhesion, and migration of leukocytes, and fibrin balance [18]. An intact and healthy endothelium expresses various anticoagulants and prevents leukocyte activation through the secreted nitric oxide (NO). Conversely, a dysfunctional endothelium shifts towards a procoagulant state through the secretion of a vasoconstrictor factor and the recruitment of immune cells, leading to an inflammatory status [19].

Numerous reports highlighted that SARS-CoV-2 could affect the endothelium of capillaries [20,21]. First, the angiotensin-converting enzyme 2 (ACE2) receptor, essential for the uptake of SARS-CoV-2 by host cells, is highly expressed on ECs. Second, viral particles included in dead ECs of different organs were found, suggesting a direct viral attack on ECs [22]. Thus, endothelial involvement and pre-existing conditions that predispose to endothelial dysfunction, such as diabetes, obesity, dyslipidaemia, smoking and disturbed blood flow, can contribute to enhanced inflammatory response and procoagulant state [23].

Coagulation abnormalities were described in acute COVID-19 [24]. At intensive care unit (ICU) admission, whole-blood thromboelastometry profiles were characterized by an acceleration of the propagation phase of blood clot formation and significantly higher clot strength [25]. Moreover, in such critical patients, high thrombin generation and impaired fibrinolysis were found [26,27].

## 3. Sustained Prothrombotic Changes in COVID-19

Emerging reports suggest that the symptoms of COVID-19 may persist beyond the acute setting [28]. Although the mechanism is still unclear, several sequelae have been reported, such as respiratory symptoms, nervous system diseases, gastrointestinal disorders, and cardiovascular conditions [28,29,30]. Notably, arterial thromboembolism (ATE) and VTE were reported in discharged COVID-19 patients [31]. After 4 months of acute illness, a hypercoagulable state was demonstrated by von Meijenfeldt et al. [32]. They found significantly elevated plasma levels of factor VIII, plasminogen activator inhibitor 1 (PAI-1) and a slight increase in von Willerbrand factor (vWF). Moreover, ex vivo thrombin-generating potential assessed by thrombomodulin-modified calibrated automated thrombinography was markedly elevated at follow-up, therefore evidencing enhanced thrombin-generating capacity and decreased plasma fibrinolytic potential [32]. Similar findings were described by other authors who also documented increased levels of EC biomarkers such as vWF, factor VIII and plasma soluble thrombomodulin [33]. Therefore, in convalescent COVID-19 patients a sustained epitheliopathy may be an important contributor to long COVID pathogenesis.

It should be emphasized that such a prolonged hypercoagulable state is not clinically relevant for the overall COVID 19 population; indeed, the overall incidence of symptomatic VTE in the unselected population is quite low (less than 3%) [31,34,35]. However, some clinical features such as a history of VTE, peak D-dimer > 3 µg/mL and predischarge C-reactive protein > 10 mg/dL were found to increase the thrombotic risk [31]. 

## 4. The Role of Heparin in COVID-19 

Anticoagulation is considered the main function of heparin; it also carries other biological actions: anti-inflammatory, anti-apoptosis and anti-cancer, the so-called pleiotropic effects of heparin [36,37,38]. 

It was hypothesized that heparin binds with inflammatory cytokines, inhibits neutrophil chemotaxis and leukocyte migration and neutralizes the complement factor C5a. Heparins may also have a role in inflammation and cellular homeostasis [39,40]. 

Heparins could protect ECs from damage through their effects on histone methylation and through regulation of mitogen-activated protein kinase (MAPK) and nuclear factor kappa-light-chain-enhancer of activated B cells (NF-κB) signalling pathway [41]. 

The endothelial glycocalyx, a proteoglycan and glycoprotein-rich layer covering the luminal side of ECs contributes to vascular homeostasis.

Glycocalyx is an extracellular structure that covers the tissue surface; it is mainly formed by heparan sulfate (HS) along with other glycoproteins and proteoglycans and constitutes a luminal mesh allowing ECs to bind soluble proteins [42]. Many viruses, such as human immunodeficiency virus, dengue virus and rabies virus utilize HS to enter their target cells [43,44,45]. Moreover, coronaviruses, including SARS-CoV-2, bind to HS to increase the virus density on the cell’s surface, thus facilitating their interaction with ACE2 receptors [46]. Recently, it was demonstrated that the spike glycoprotein’s receptor-binding domain firstly interacts with HS, contributing to a conformational change in the protein. This change allows the binding of the virus to the ACE2 receptor [47]. Unfractionated Heparin (UFH) may compete with SARS-CoV-2 for the binding to HS, inhibiting the virus attachment to the cell surface and the viral entry. Some authors also demonstrated that, in vitro, 0.3–0.7 U/mL of UFH can reduce the percentage of SARS-CoV-2 infected cells. The binding of heparin/HS to S trimers enhances the binding to ACE2, likely increasing multivalent interactions with the target cells. Clausen et al. revealed HS as a novel attachment factor for SARS-CoV-2 and suggested the possibility of using HS mimetics, HS degrading lyases, and metabolic inhibitors of HS biosynthesis against COVID-19 [47]. 

Main protease (M^pro^) is a key enzyme of coronavirus that plays an essential role in concerning viral replication and transcription. Li et al. demonstrated that heparin binds to SARS-CoV-2 M^pro^, inhibiting its proteolytic activity in vitro. Therefore, heparin might inhibit SARS-CoV-2 replication and transcription by inhibiting the activity of the SARS-CoV-2 M^pro^ protein [48]. 

Potje et al. showed that plasma from hospitalized COVID-19 patients contained increased levels of glycocalyx components and increased heparanase activity, indicating glycocalyx disruption. Moreover, plasma from COVID-19 patients also resulted in glycocalyx shedding and disturbed redox balance in healthy ECs of the umbilical veins cells [49]. Low-molecular-weight heparins (LMWH) inhibited glycocalyx perturbation induced by plasma from COVID-19 patients [49]. 

Another intriguing potential therapeutic role of heparin in COVID-19 seems to be the inhibition of heparanase, an endothelial glycocalyx-degrading enzyme that contributes to vascular leakage and inflammation. The activity of heparanase was associated with disease severity in COVID-19 patients, and Buijsers et al. demonstrated that LMWH could reduce its activity [50]. 

The postulated therapeutic effects of heparin are summarized in Figure 1.

## 5. Evidence of Heparin Use in COVID-19 Acute Illness

We selected randomized trials and meta-analyses by searching the terms “heparin and COVID 19” in the Pubmed database. From 1397 results, we only selected randomized trials and meta-analyses and we found 36 results. According to the purpose of our research, we selected 11 randomized trials (Table 1) and 5 meta-analyses with heparin therapy, mostly LMWH. We excluded 20 studies because they were off-topic. We decided to exclude retrospective studies because of their intrinsic limitations (i.e., small sample size, selection and confounding bias).

In the HESACOVID Trial, Lemos et al. showed in a small group of patients with severe disease (10 patients per arm) that 14 days of therapeutic anticoagulant dose (enoxaparin or UFH) versus thromboprophylaxis dose (enoxaparin or UFH) significantly reduced the need for mechanical ventilation and improved blood gas parameters. Thrombocytopenia was not reported, and only two minor bleeding were observed in the therapeutical dose group [51].

The INSPIRATION trial compared the use of enoxaparin at the intermediate dose (1 mg/kg/daily) versus prophylactic dosage (enoxaparin, 40 mg daily) in 600 patients admitted to the ICU with COVID-19. After 30 days of continued therapy, no statistical difference in outcomes (VTE or ATE, all cause-death) was found among treatments (45.7% in the intermediate dose group and 44.1% in the prophylaxis group; absolute risk difference, 1.5% [95% confidence interval (CI), −6.6% to 9.8%]; odds ratio (OR), 1.06 [95% CI, 0.76–1.48]; *p* = 0.70). Interestingly, a trend towards a better primary composite outcome with the use of a prophylactic dosage of heparin was found in women (36.6% prophylactic vs. 47.4% intermediate, *p* = 0.06). Major bleeding events were observed in 2.5% of patients in the intermediate group vs. 1.4% of those in the prophylactic group, not meeting the noninferiority criteria (*p* for noninferiority > 0.99); moreover, six cases of thrombocytopenia were reported the intermediate-dose group. Therefore, the authors suggested to avoiding the routine empirical use of intermediate-dose prophylactic anticoagulation in unselected patients admitted to the ICU [52]. The authors extended the follow-up to 90 days, confirming that there was no difference between the two groups in the primary composite outcome. New bleeding events did not occur, and cases of thrombocytopenia were not described [53].

The multicentre, randomized AntiCoagulaTlon cOroNavirus (ACTION) trial enrolled about 600 patients and was randomized in a 1:1 fashion therapeutic or prophylactic anticoagulation. The in-hospital therapeutic anticoagulation protocol was the following: rivaroxaban (20 mg or 15 mg daily) for stable patients, or initial subcutaneous enoxaparin (1 mg/kg twice per day) or intravenous UFH for clinically unstable patients, followed by rivaroxaban until day 30. Prophylactic anticoagulation was the standard dosage of enoxaparin or unfractionated heparin, administered only during the hospitalization. As a results, the therapeutic dose showed no significant benefit over the prophylactic one in primary outcomes (mortality, length of hospitalization, or duration of oxygen therapy after 30 days, respectively, 34.8% vs. 41.3%—*p* = 0.40), while a statistically significant increase in major or clinically relevant non-major bleeding was reported in the therapeutic group (8% vs. 2%, *p* = 0.0010). Notably, no thrombocytopenia was reported. Based on these data, the authors suggest avoiding the use of direct oral anticoagulants (DOACs) unless previously practiced for known indications [54]. 

In an open-label, adaptive, multiplatform, randomized clinical trial that included three international adaptive platform trials (REMA-CAP, ACTIV-4a, ATTACC) on critically ill patients, 534 patients were assigned to therapeutic-dose anticoagulation while 564 were assigned to thromboprophylaxis heparin dose during the hospitalization. The therapeutic dose of heparin showed no significant advantage in reducing mortality compared to prophylactic therapy (62.7% and 64.5%, respectively; adjusted OR 0.84; 95% Credible Interval, 0.64 to 1.11). Major bleeding rates were slightly but not significantly increased in the therapeutic dose group (3.8% vs. 2.3%, adjusted OR 1.48–95% Credible Interval 0.75 to 3.04). Thrombocytopenia was not reported [55].

Such investigators conducted a same fashion multiplatform trial that enrolled 2219 non critically ill patients. Conversely, a significant reduction in mortality and disease intensity with the use of anticoagulant dose compared to prophylactic therapy was observed but with a higher rate of major bleeding (1.9% vs. 0.9%). Thrombocytopenia was not reported [56].

Perepu et al. conducted a 1:1 randomized study in 176 patients comparing enoxaparin at an intermediate dose (1 mg/kg or 0.5 mg/kg twice daily if the body mass index was ≥30) with standard prophylactic therapy in patients with severe COVID-19 (ICU admission and/or evidence of coagulopathy). Enoxaparin dose was administered until hospital discharge or occurrence of a clinical event. No difference was found between the two dosages for the primary outcome (30-day mortality for any cause) that occurred in 15% of those who underwent the therapeutic dose and 21% treated with the prophylactic dose (OR 0.66; 95% CI, 0.30–1.45; *p* = 0.31). Major bleeding was comparable (2% of patients in each arm), and thrombocytopenia was not reported [57].

RAPID was an open-label, multicentre randomized trial, that enrolled 465 patients with elevated d-dimer values comparing the standard prophylactic heparin dose (enoxaparin 4000 UI twice daily if BMI > 40) with the standard therapeutic dose. Treatment was continued until hospital discharge, or day 28, or study withdrawal, or death. At 28 days, the therapeutic dose was not significantly associated with a reduction in the primary outcome (death, invasive mechanical ventilation, non-invasive mechanical ventilation, or admission to ICU), but significantly fewer deaths occurred (1.8% vs. 7.6%, *p* = 0.006). Major bleeding events did not significantly differ in the two groups (0.9% in the therapeutic group and 1.7% in the prophylactic one, *p* = 0.69). No cases of thrombocytopenia were signalled [58]. 

The multicentre randomized trial HEP-COVID enrolled 257 patients with D-dimer greater than four times the upper limit and with Sepsis-Induced Coagulopathy (SIC) score ≥ 4 and compared the effects of standard anticoagulant dose heparin therapy vs. standard prophylactic dose or standard intermediate therapy during a hospital stay. The primary outcome was VTE, ATE or death from any cause. Therapeutic dose was associated with a reduction in thromboembolic events and death at 28 days (28.7% vs. 41.9%, relative risk (RR), 0.68; 95% CI, 0.49–0.96; *p* = 0.03), although this benefit was not observed in ICU patients (51.1% vs. 55.3%; RR 0.92; 95% CI, 0.62–1.39; *p* = 0.71). Higher but not significant rate of major bleeding occurred with therapeutic dose (4.7% vs. 1.6%, RR 2.88; 95% CI, 0.59–14.02; *p* = 0.17). Only one case of thrombocytopenia was reported [59]. 

The BEMICOP Study, an open-label, randomized control trial, compared the therapeutic dose of bemiparin (115 IU/kg daily) versus prophylaxis (bemiparin 3500 IU daily), for 10 days in COVID-19 patients hospitalized with non-severe pneumonia but elevated D-dimer. A total of 65 patients were included in the primary analysis to assess the primary efficacy outcome (a composite of death, intensive care unit admission, need for mechanical ventilation support, development of moderate/severe acute respiratory distress, and VTE or ATE). The use of the therapeutic bemiparin dose did not improve the outcome as events occurred in 22% of the therapeutic dose group and in 18% of the prophylactic dose group (absolute risk difference 3.6%; 95% CI, −16–24%; OR 1.26; 95% CI, 0.37–4.26; *p* = 0.95). No major bleeding was registered, and thrombocytopenia was not assessed [60]. 

In the X-COVID-19, a multicentre, open-label, randomized trial, the intermediate dose of enoxaparin (40 mg twice daily) was compared with the standard prophylactic dose. The study was interrupted prematurely due to slow recruitment. Treatment duration was 7 days in the prophylactic dose group and 9 days in the therapeutic dose group. Although underpowered, the analysis of 183 patients showed no deep vein thrombosis in COVID-19 hospitalized patients in both groups; however pulmonary embolism was observed only in the prophylactic dose group (six patients). No major bleeding and thrombocytopenia events were reported [61]. 

A meta-analysis conducted by Kow et al. analyzed the data from these randomized trials. The authors acknowledged the biases reported in single studies. Statistical analysis showed no differences in mortality in the therapeutic/intermediate-dose groups versus the prophylactic dose group. A benefit in reducing VTE events was observed in the subgroup of patients with severe COVID-19 without a significant increase in bleeding. The results suggested a benefit of anticoagulant therapy over the prophylactic dose. The failure to reduce mortality in severe patients seemed to be due to a delayed start of therapy. No trial evaluated the risk of individual bleeding; therefore, this could explain the observed increase in major bleeding [62].

We also found meta-analyses of non-randomized trials. 

Hasan et al. included 12 studies on patients admitted to the ICU and highlighted a high prevalence of thromboprophylaxis failure; therefore, they suggested an individualized approach rather than a fixed prophylactic dose [63].

A second meta-analysis on 11 studies was conducted by Sridharan et al. [64] in hospitalized COVID-19 adult patients and confirmed an increased rate of VTE in those patients admitted to the ICU. The authors showed a lower incidence of VTE in all hospitalized patients with the use of the therapeutic dose of anticoagulation compared to the prophylactic one, although an increase in major bleeding was observed.

A third meta-analysis that investigated the impact on mortality using the prophylactic heparin dose was performed by Abdel-Maboud et al. Authors included, in the final analysis, five studies and showed a positive effect of prophylactic dose only in patients with moderate symptoms and a combined D-dimer  >  3 µg/L, a platelet count  >  100 × 10^9^/L, and a prothrombin time  <  14 s, regardless of comorbidity, sex or age [65].

Parisi et al. analyzed 29 studies with very heterogeneous data both on the type of anticoagulant used and on the dosage. Despite the limitations of the examined sample, there was a reduction in mortality in the group that underwent anticoagulant therapy; however, an increased percentage of bleeding was observed. The authors suggested that it is advisable to use prophylactic doses in non-severe diseases [66].

## 6. Postdischarge Prophylaxis with Heparin

As discussed above, the risk of ATE and VTE in patients with COVID-19 extends beyond their hospitalization. The role of heparin for postdischarge prophylaxis was investigated by some authors. The COVID-19 Research Consortium of our health system

(CORE-19) registry assessed 90-day postdischarge VTE and ATE: the use of anticoagulants, mostly prophylactic doses, was associated with a 46% decrease in VTE, but the subgroup of patients who received heparin thromboprophylaxis was small as enoxaparin was used only in 1.3% of patients [67]. 

In a retrospective analysis conducted in hospitalized patients with symptomatic COVID-19 infection who were discharged, no VTE events occurred in those who received extended prophylaxis; however, enoxaparin was used only in 12.9% of cases [68]. To date, no randomized trial that assesses the role of heparin for VTE prevention in COVID-19 patients after the acute illness has been published.

## 7. Discussion

Data from several studies confirmed the increased incidence of VTE in COVID-19 patients and its impact on mortality, especially in ICU patients. An increase in bleeding was also observed [69,70,71]. The studies we discussed [51,52,53,54,55,56,57,58,59,60,61] and the available meta-analyses [62,63,64,65,66] revealed the need for a tailored anticoagulant therapy. Heparin was shown to reduce mortality both at prophylactic and therapeutic dosages; however, an increase in bleeding was reported [72]. An interesting issue is an outcome according to sex difference, as a higher rate of hospitalizations, ICU admissions and death were reported in men [73]. In almost all randomized trials a higher prevalence of males was observed; only in two trials a subgroup analysis according to sex was performed [52,58]; however, no significant difference was found among treatments dosages according to sex, although a trend towards a better outcome with the use prophylactic dose of heparin was described in women [52].

Although immune thrombocytopenia has emerged as a complication of COVID-19 [74], low platelets count observed in critically ill patients may be related to several clinical conditions such as heparin-induced thrombocytopenia or sepsis-induced coagulation. In the analyzed trials, thrombocytopenia was rarely reported.

The American Society of Hematology (ASH) guideline panel suggested the prophylactic versus intermediate or therapeutic dosage for patients with acute or critical COVID-19 illness without VTE (conditional recommendation based on very low certainty in the evidence about effects) [75], and a subsequent update of the guidelines has been kept the same recommendation but with improved data (conditional recommendation based on low certainty in the evidence about effects) [76]. The update of the National Institutes of Health (NIH) guidelines limited the use of therapeutic heparin dosage to hospitalized patients who require low oxygen levels, are not pregnant and with high dimer values (strength of recommendation: weak; Quality of Evidence (QoE): IIa), while the prophylactic dose should be used, unless contraindicated, in patients with more severe forms (strength of recommendation: strong; QoE: I), except in case of VTE diagnosis [77]. A flow-chart based on available evidence is suggested in Figure 2. 

The impact of other anticoagulants in improving the outcome of COVID-19 has been evaluated. In two studies the prophylactic dose of Fondaparinux (2.5 mg daily) showed similar efficacy and safety compared to enoxaparin prophylactic dosage [78,79]. Conversely, in another study the rate of major or clinically relevant bleeding was significantly higher in patients treated with fondaparinux [80].

Also, the use of DOAC in COVID-19 was investigated. A recent meta-analysis of 12 studies involving 30,646 patients concluded that previous DOAC therapy at time of COVID-19 diagnosis did not improve clinical outcomes [81]. Furthermore, a randomised trial did not demonstrate an impact on disease progression of the prophylactic dose of rivaroxaban (10 mg) in high-risk adults with mild COVID-19 [82]. Lack of improvement in patients’ outcome with the use of DOAC is probably related to a different pathogenesis of VTE in COVID-19 [83].

Postdischarge thromboprophylaxis is not recommended, although it could be considered in patients at high risk of VTE and low risk of bleeding [84]. Only the data from the CORE-19 registry seem to confirm a reduction in VTE in patients receiving postdischarge thromboprophylactic therapy [67]; conversely, other studies did not show significant differences in terms of VTE and bleeding occurrence regardless of the prophylaxis with heparin after discharge [85,86]. However, several authors recognized some characteristics that identify patients in whom prophylaxis is advisable such as the history of VTE, peak D-dimer > 3 µg/mL and predischarge C-reactive protein >10 mg/dL, International Medical Prevention Registry on Venous Thromboembolism (IMPROVE) VTE score ≥ 4 [87,88]. The optimal duration of thromboprophylaxis is unknown. In the Medically Ill hospitalized Patients for COVID THrombosis Extended ProphyLaxis with rivaroxaban ThErapy (MICHELLE) study, an open-label, multicentre, randomized trial, the safety and efficacy of rivaroxaban 10 mg in a group of postdischarge COVID-19 patients was evaluated. In patients at high risk, thromboprophylaxis with rivaroxaban 10 mg/day for 35 days improved clinical outcomes compared with no extended thromboprophylaxis [88]. 

### Limitations

The main limitation of the trials is due to the pandemic scenario in which recruitment and follow-up were challenging in several countries. Therefore, most of the studies included a limited number of patients with a consequent reduction of statistical power.

Second, the criteria used to classify patients into critically ill or non-critically ill differed across the studies. Third, the short duration of study treatment limits the evaluation of long-term impact. Finally, the intrinsic limitation of retrospective studies included in some meta-analyses encompass selection bias and confounding bias.

## 8. Conclusions

Heparin plays a central role in COVID-19 treatment as demonstrated in several randomized clinical trials, although with some limits due to the COVID-19 related pandemic. The appropriate dose of heparin is still being debated and a reasoned approach according to clinical scenarios is necessary. According to the most updated guidelines, the prophylactic dose is recommended in all hospitalised patients unless contraindicated, while the therapeutic dose could be considered only in non-pregnant patients requiring low-flow supplemental oxygen, with increased D-dimer levels and low bleeding risk. In case of thrombocytopenia, the prophylactic dose of fondaparinux could be considered. In patients previously treated with DOAC for underlying conditions, these medications should be continued after a diagnosis of COVID-19.

Heparin may also have a role in the prevention of postdischarge COVID-19 sequelae in the presence of high-risk clinical features that raise the risk of thrombotic complications.

## Figures and Tables

**Figure 1 jcm-11-03099-f001:**
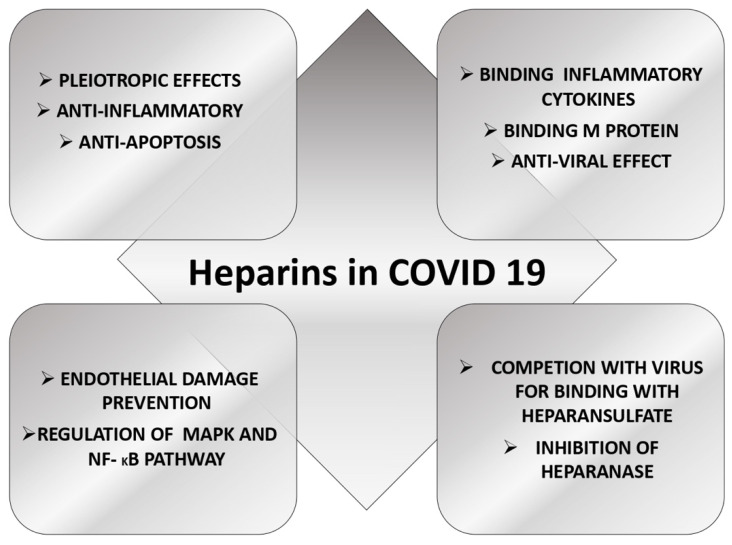
Potential therapeutic effects of heparin in COVID 19.

**Figure 2 jcm-11-03099-f002:**
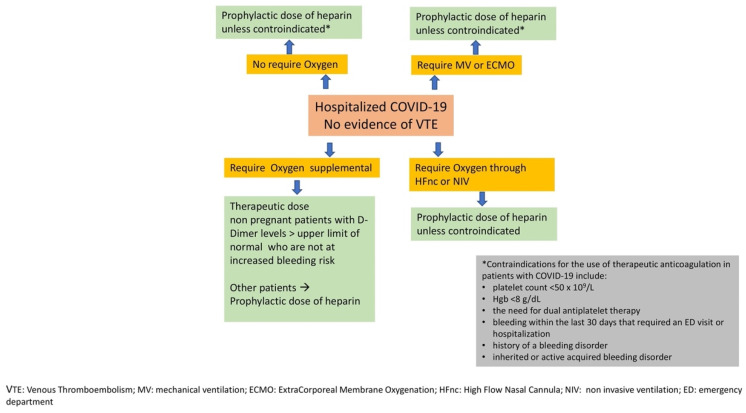
Flow-chart on suggested heparin dosage according to different clinical scenarios.

**Table 1 jcm-11-03099-t001:** Randomized trials on heparin in COVID-19.

TRIAL	Methods	Mean Age/Male	Interventions	Results
HESACOVID [51] REBEC RBR-949z6v PHASE 2	20 PTS, RCT, OL, LMWH	Td: 55 years/90% Pd: 58 years/70%	Td vs. Pd	Td reduces the need for mechanical ventilation and improves blood gas parameters
INSPIRATION [52,53] NCT04486508 PHASE 3	562 PTS, RCT, OL, LMWH	Id: 62 years/58.7% Pd: 61 years/57%	Id vs. Pd	No difference in the 30-day outcomes (ICU) No difference in the 90-days outcomes (ICU)
ACTION [54] NCT04394377 PHASE 4	615 PTS, RCT, OL, LMWH, UH, DOAC	Td: 56.7 years/62% Pd: 56.5 years/58%	Td vs. Pd	No difference in primary outcome between Td and Pd Bleeding increased statistically with Td (DOAC)
The REMAP-CAP/ACTIV-4a/ATTACC trial (severe) [55] NCT02735707 PHASE 3, NCT04505774 PHASE 4, NCT04359277 PHASE 4, NCT04372589 PHASE 2/3	1098 PTS, RCT, OL, LMWH, UH	Td: 60.4 years/72.2% Pd: 61.7 years/67.9%	Td vs. Pd	No difference in mortality
The REMAP-CAP/ACTIV-4a/ATTACC trial (moderate) [56] NCT02735707 PHASE 3, NCT04505774 PHASE 4, NCT04359277 PHASE 4, NCT04372589 PHASE 2/3	2131 PTS, RCT, OL, LMWH, UH	Td: 59 years/60.4% Pd: 58.8 years/56.9%	Td vs. Pd	Reduction in mortality and disease intensity with Td
Perepu et al. [57] NCT04360824 PHASE 4	173PTS, RCT, OL, LMWH	Id: 65 years/54% Pd: 63.5 years/58%	Id vs. Pd	No difference in ICU patients
RAPID [58] NCT04362085 PHASE 3	465 PTS, RCT, OL, LMWH, UH	Td: 60.4 years/53.9% Pd: 59.6 years/59.5%	Td vs. Pd	28 days mortality reduction with Td in moderately ill patients
HEP-COVID [59] NCT04401293 PHASE 3	253 PTS, RCT, DB, LMWH, UH	Td: 65.8 years/52.7% Pd: 67.7 years/54.8%	Td vs. Pd	30 days reduction in thromboembolic events and death with Td in moderately ill patients
BEMICOP STUDY [60] NCT04604327 PHASE 3	65 PTS, RCT, OL, LMWH	Td: 63 years/53.1% Pd: 62.3 years/72.7%	Td vs. Pd	Td does improve clinical outcomes
X-COVID 19 [61] NCT04366960 PHASE 3	183 PTS, RCT, OL, LMWH	Id: 60 years/61.5% Pd: 59 years/64.1%	Id vs. Pd	No DVT in both groups; 6 vs. 0 PE in Pd group

PTS: patients; RCT: randomizes control trial; OL: open label; DB: double blind; LMWH: low molecular weight heparin; UH: unfractionated heparin; DOAC: direct-acting oral anticoagulants; Td: therapeutic dose; Id: intermediate dose; Pd: prophylactic dose; DVT: deep vein thrombosis; PE: pulmonary embolism.

## Data Availability

Data sharing not applicable.

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
