# Peer review of "The Role of Heparin in COVID-19: An Update after Two Years of Pandemics"

_jcm, 2022, doi:10.3390/jcm11113099_

Round 1
Reviewer 1 Report
The article under review represents an investigation of available evidence in regards to heparin treatment in COVID-19 patients. The authors conclude that although the optimal dose of heparin is still debated, heparin plays a central role in COVID-19 treatment. The article is well-written and covers an important characteristic of COVID-19 therapy. However, the following aspects should be addressed:
- In the introduction, the authors write: “Furthermore, many patients showed coagulation abnormalities: increased D-dimer concentration, a decrease in platelet count, and a prolongation of the prothrombin time suggested the presence of a hypercoagulable state in COVID-19 that could lead to an increased risk of thromboembolic complications [6]” It would be beneficial to also include other recently published trials investigating thromboembolic complications in terminally ill COVID-19 patients such as https://doi.org/10.1101/2021.12.27.21268264
- Furthermore, please clarify in the introduction how the current review article differentiates itself from other published review articles on heparin as a COVID-19 treatment strategy (such as PMID: 32519894)
- In section 5, please elaborate further upon what the study inclusion criteria were.
- In the conclusion section, it is stated: “Heparin plays a central role in COVID-19 treatment as demonstrated in several randomized clinical trials although with some limits due to the COVID-19 related pandemic.” It would be beneficial to briefly mention what the most important limits are.
- In figure 1, the word “Heparins” appears to have a spell-checking red underline. Please remove it.
- Increase the figure resolution as it is currently difficult to read the text.
Author Response
The article under review represents an investigation of available evidence in regards to heparin treatment in COVID-19 patients. The authors conclude that although the optimal dose of heparin is still debated, heparin plays a central role in COVID-19 treatment. The article is well-written and covers an important characteristic of COVID-19 therapy. However, the following aspects should be addressed:
Authors: We would like to thank the reviewer for his valuable comments that allowed us to improve our manuscript. The following are our replies.
1) In the introduction, the authors write: “Furthermore, many patients showed coagulation abnormalities: increased D-dimer concentration, a decrease in platelet count, and a prolongation of the prothrombin time suggested the presence of a hypercoagulable state in COVID-19 that could lead to an increased risk of thromboembolic complications [6]” It would be beneficial to also include other recently published trials investigating thromboembolic complications in terminally ill COVID-19 patients such as https://doi.org/10.1101/2021.12.27.21268264
R: We have updated the references according to reviewer’s suggestions and slightly changed the text (line 37 and 41-43, references 7, 8-14).
2) Furthermore, please clarify in the introduction how the current review article differentiates itself from other published review articles on heparin as a COVID-19 treatment strategy (such as PMID: 32519894)
R: We thank the reviewer for the suggestion, we improved the introduction adding a sentence in line 104-107.
3) In section 5, please elaborate further upon what the study inclusion criteria were.
R: We specified the inclusion criteria [lines 265-431] and we added a meta-analysis [lines 924-928, REF 65].
4) In the conclusion section, it is stated: “Heparin plays a central role in COVID-19 treatment as demonstrated in several randomized clinical trials although with some limits due to the COVID-19 related pandemic.” It would be beneficial to briefly mention what the most important limits are.
R: We illustrated the limits in lines 1044-1051.
5) In figure 1, the word “Heparins” appears to have a spell-checking red underline. Please remove it.
R: We thank the reviewer for the suggestion, we removed it
6) Increase the figure resolution as it is currently difficult to read the text.
R: We improved the resolution.
Reviewer 2 Report
Dear authors,
Your review provided an overview of the use of heparin in the treatment of patients with Covid-19 infection. I found that the article is well structured. However, I have some comments:
Major comments:
- Please provide high resolution images (both Fig 1 and 2) as well as Table 1.
- The authors provided a good synthesis of the evidence of heparin in the protective effect on VTE, but what about safety of heparin in these patients? I appreciated that you reported the major bleeding events for each study, however, sometimes this information is not useful due to the lack of other information (dose, treatment duration) Moreover, also other events such as thrombocytopenia should be reported given that immune thrombocytopenia could be an important complication of covid-19. (PMID: 32984764)
- Why do you limited your analysis only to randomized clinical trials and meta-analysis? Please report the rational (line 146). Maybe observational studies could provide additional information on safety and effectiveness in specific populations e.g., elderly, nursing homes populations.
- Few studies reported different outcome between male and female to anticoagulants in COVID-19 (PMID: 34454035), does the authors found any sex differences in response to heparin? Please discuss it. This is in line with phrase 253 where you suggested the need of a tailored anticoagulant therapy.
Minor comments:
- Table 1: Please provide NCT numbers, trial phase, number of female/male and median age where available. This will give readers a better overview and understand the differences between trials at a glance.
- Other anticoagulants (e.g., novel oral anticoagulants) were tested to treat covid-19? Please discuss it reporting possible difference with the treatment with heparin.
Author Response
Your review provided an overview of the use of heparin in the treatment of patients with Covid-19 infection. I found that the article is well structured. However, I have some comments:
Authors: We would like to thank the reviewer for his valuable comments that allowed us to improve our manuscript. The following are our replies.
Major comments:
- Please provide high resolution images (both Fig 1 and 2) as well as Table 1.
R: We improved the resolution of images and table.
- The authors provided a good synthesis of the evidence of heparin in the protective effect on VTE, but what about safety of heparin in these patients? I appreciated that you reported the major bleeding events for each study, however, sometimes this information is not useful due to the lack of other information (dose, treatment duration) Moreover, also other events such as thrombocytopenia should be reported given that immune thrombocytopenia could be an important complication of covid-19. (PMID: 32984764)
- R: We reported the information suggested by the reviewer. Also, we added the reports of thrombocytopenia in the trials and this issue was added in discussion (lines 1016-1019).
- Why do you limited your analysis only to randomized clinical trials and meta-analysis? Please report the rational (line 146). Maybe observational studies could provide additional information on safety and effectiveness in specific populations e.g., elderly, nursing homes populations.
R: We thank the reviewer for the comment, we decided to include only randomized trial to reduce the risk of bias due of the retrospective study. We motivated our choice, and we specified the inclusion criteria [lines 265-431]. However, we reported a comprehensive metanalysis of 29 retrospective studies [R. Parisi et al., ref 66].
- Few studies reported different outcome between male and female to anticoagulants in COVID-19 (PMID: 34454035), does the authors found any sex differences in response to heparin? Please discuss it. This is in line with phrase 253 where you suggested the need of a tailored anticoagulant therapy.
R: We thank the reviewer for the comment. In almost all randomized trial, a higher prevalence of male was observed; only in two trial a subgroup analysis according to gender was performed (ref 52, 58), however, no significant difference was found among treatments dosages according to sex, although a trend towards a better outcome with the use prophylactic dose of heparin was described in women [ref 52]. We discussed this issue in discussion section [lines 956-962].
Minor comments:
- Table 1: Please provide NCT numbers, trial phase, number of female/male and median age where available. This will give readers a better overview and understand the differences between trials at a glance.
R: We thank the reviewer for his valuable comments, we improved the table as suggested.
- Other anticoagulants (e.g., novel oral anticoagulants) were tested to treat covid-19? Please discuss it reporting possible difference with the treatment with heparin.
R: Other anticoagulants were tested in Covid-19 treatment such as Fondaparinux or DOACs. However, a comprehensive analysis of such data and the discussion about the differences with heparin merits a new dedicated chapter. Although the topic is quite interesting, we retain that is beyond the scope of our paper.
Round 2
Reviewer 2 Report
Dear authors,
Thank you for answer my comments.
In the submitted manuscript there are comments from the authors [MOU1] as well as space between chapters. Moreover figure legends were eliminated. I suggest to carefully review the editing before final submission.
Author Response
We revised the editing, we wish to thank the reviewer for the suggestions.